# SymNet 2.0: Effectively handling Non-Fluents and Actions in Generalized Neural Policies for RDDL Relational MDPs

**Vishal Sharma**[1]     **Daman Arora**[1]     **Florian Geißer**[2]     **Mausam**[1]     **Parag Singla**[1]

[1]Indian Institute of Technology Delhi {vishal.sharma, cs5180404, mausam, parags}@cse.iitd.ac.in
[2]Independent Reseacher, {florian.geisser.work}@gmail.com

## Abstract

Relational MDPs (RMDPs) compactly represent an infinite set of MDPs with an unbounded number of objects. Solving an RMDP requires a *generalized* policy that applies to all instances of a domain. Recently, Garg et al. proposed SymNet for this task – it constructs a graph neural network that shares parameters across all instances in a domain, thus making it applicable to any instance in a zero-shot manner. Our analysis of SymNet reveals that it performs no better than random on 1/4th of planning competition domains. The key reasons are its design choices: it misses important information during graph construction, leading to (1) poor generalizability, and (2) potential non-identifiability of different actions.

In response, our solution, SYMNET2.0, substantially augments SymNet's graph construction approach by introducing additional nodes and edges which allow a better transfer of important information about a domain. It also improves SymNet's action decoders with relevant information from objects to make different actions identifiable during scoring. Extensive experiments on twelve competition domains, where we use imitation learning over data generated from the PROST planner, demonstrate that SYMNET2.0 performs vastly better than SymNet. Interestingly, even though SYMNET2.0 is trained over data from PROST, it outperforms the planner on several test instances due to former's ability to scale to large instances in a zero-shot manner.

## 1 INTRODUCTION

A Relational Markov Decision Process (RMDP) (Boutilier et al. [2001]) is a first-order representation of a planning domain usually represented in a description language like the Probabilistic Planning Domain Definition Language (PPDDL) [Younes et al., 2005] or the Relational dynamic influence diagram language (RDDL) [Sanner, 2010]. Finding solvers for an RMDP which perform well on any instance of a domain has been a long-standing goal of AI planning research. Motivated by the recent progress in deep neural models, multiple works [Groshev et al., 2018, Toyer et al., 2018, Garg et al., 2019, 2020, Ståhlberg et al., 2022] learn generalized neural reactive policies, which are trained on a set of (smaller) training instances, and can be transferred to a set of (larger) test instances in a zero-shot manner. Our focus is on learning generalized neural policies for RMDPs expressed in RDDL, where SymNet [Garg et al., 2020] has demonstrated initial feasibility.

However, our analysis reveals that SymNet performs no better than random on 1/4th of the domains of the International Probabilistic Planning Competition[1] (IPPC 2011 and 2014), and even in several others where it seemingly does well, it performs significantly worse than PROST [Keller and Eyerich, 2012], the state-of-the-art *online* planner for RDDL RDMPs. This points to a significant research gap between what is possible, and what is currently achievable. In this paper, our goal is to examine whether we can fill this gap by a better design of the underlying neural architecture.

At a high level, SymNet compiles an RMDP instance to an *instance graph*, with nodes representing object tuples, and edges representing connections in the Dynamic Bayes Net (DBN) corresponding to the instance. Given a state, a Graph Attention Network [Veličković et al., 2018], on top of the instance graph, computes embeddings for each node. A subset of these nodes embeddings (or their aggregate) is then passed through an action decoder network to output a score for the ground actions. The network is typically trained using a loss function based on reinforcement learning (RL).

We identify two key challenges with SymNet's design choices. First, its handling of *non-fluents*, variables which

---

[1]https://www.icaps-conference.org/competitions/

*Accepted for the 38th Conference on Uncertainty in Artificial Intelligence* (UAI 2022).

are static throughout the application of a policy but whose value depends on the given instance, is somewhat ad-hoc. Many non-fluents do not directly correspond to specific nodes in the graph, instead they are compiled away. This leads to a significant problem with generalizability of the network to instances where the value of those non-fluents differs. Second, the action decoder for a ground action takes an aggregation over as input those node embeddings that are affected by the action; it does not necessarily take all the objects that are arguments of the action. This can lead to a problem of action non-identifiability: two ground actions with different object arguments affecting the exact same set of objects get exactly the same score. We describe these in detail through a running example in Section 3.2.

To mitigate these issues we present SYMNET2.0[2], which substantially augments SymNet's architecture. To handle non-fluents in a principled manner, SYMNET2.0's instance graph creates a node for each object tuple appearing as an argument to any non-fluent. In order to connect these nodes to the rest of the network it additionally creates singleton nodes for each object in the instance. These singleton object nodes connect to all object-tuple nodes that contain this object. To handle action non-identifiability during decoding, we additionally pass the embeddings of all singleton nodes that appear as action arguments in the action.

We train both SymNet and SYMNET2.0 with imitation learning on a dataset generated by planning using PROST on training instances; this helps us circumvent the training and exploration issues faced by RL algorithms. Extensive experiments on twelve IPPC domains demonstrate that SYM-NET2.0 performs vastly better than SymNet, obtaining a gain of more than 40% relative performance on half of the domains, and a gain of approx. 50% relative performance in the aggregate metric. We perform further studies by analyzing specific domains to characterize the various settings in which SYMNET2.0 outperforms SymNet. Interestingly, though SYMNET2.0 uses data generated from PROST, due to its offline nature, which requires only a forward pass during inference, SYMNET2.0 outperforms PROST on large instances of several domains; in some cases by a significant margin. This opens up new avenues for exciting research that combines online planners with policies learned using neural models.

## 2 BACKGROUND AND RELATED WORK

### 2.1 RELATIONAL MDPS AND RDDL

A Relational Markov Decision Process (RMDP) [Boutilier et al., 2001] domain, denoted by $R_M$, represents a factored MDP in a first order form as a tuple $(C, SP, A, \mathcal{O}, T, R, H, s_0, \gamma)$, where $SP$ and $A$ denotes the

---

set of state, respectively, action predicates; $\mathcal{O}$ denotes the set of objects, where each object is associated with a class type in $C$. The set of transition functions is denoted by $T$, the set of reward functions by $R$. Additionally, $H$ denotes the finite horizon and $\gamma$ the discount factor. Replacing the arguments of a predicate with an object-tuple of type-consistent objects is called grounding the predicate. Grounding the predicates of $SP$ results in a set of state-variables, denoted by $SP_{\mathcal{O}}$, and grounding the predicates of $A$ results in a set of ground actions, denoted by $A_{\mathcal{O}}$. An assignment to all $SP_{\mathcal{O}}$ denotes a state $s \in \mathcal{PS}(SP_{\mathcal{O}})$ where $\mathcal{PS}$ denotes the power set. The initial state is denoted by $s_0$.

The Relational Dynamic Influence Diagram Language (RDDL) [Sanner, 2010] represents an RMDP using two components: 1) a domain description provides predicates $SP$ and $A$, object types $C$, as well as first-order transition and reward functions $T$ and $R$; and 2) an instance description specifies ground objects $\mathcal{O}$, initial state $s_0$, as well as horizon $H$ and discount factor $\gamma$. Furthermore, the set of state predicates ($SP$) is divided into state-fluents ($SF$) and non-fluents ($NF$), where the former are predicates where the assignment of induced ground variables can change over time, and the latter are predicates whose ground variables' assignment remains static. Note that two instances induced by the same domain can have different assignments of ground variables induced by $NF$. We denote with $O_{SF}$ and $O_{NF}$ the set of object tuples that appear in $SF$, respectively $NF$. Given an RDDL instance, its transition semantics can be represented in the form of a Dynamic Bayesian Network (DBN) capturing dependencies among state-variables and ground actions [Mausam and Kolobov, 2012].

### 2.2 TRANSFER LEARNING FOR RMDPS

We define the problem of *Transfer Learning for RMDPs (TLR)* as follows. Given an RMDP $R_M$ and a set of instances of $R_M$ expressed in RDDL, the goal of TLR is to learn a generalized neural network $\mathcal{N}(I)$ parameterized by instance $I$, with a (tied) set of weight parameters $w$ independent of $I$, such that $\mathcal{N}(I)$ takes as input a state $s$ of instance $I$, and outputs a distribution over actions in the action space of $I$, i.e. $\mathcal{N}(I) : \mathcal{PS}(SP_O) \to p(A_O)$ where $p(A_O)$ represents a probability distribution over all ground actions $A_O$. We study this problem in the *offline planning* setting, i.e., at execution time, the action in a given state may be identified with minimal computation (e.g., table lookup or a forward pass), as opposed to a deliberative lookahead search, as in online planning.

### 2.3 RELATED APPROACHES

Offline planning in MDPs is a well-studied problem, e.g., Labeled RTDP [Bonet and Geffner, 2003], HMDPP [Keyder and Geffner, 2008], ReTrASE [Kolobov, 2009], Glut-

---

[2]Code released at https://github.com/dair-iitd/symnet2

ton [Kolobov et al., 2012]. Generalized planning for Relational MDPs also has a long history, with early work trying to construct features that can transfer across instances [Fern et al., 2003, Guestrin et al., 2003, Mausam and Weld, 2003, Natarajan et al., 2011]. Recent work has studied generalized planning for building fully observable non-deterministic planners (FOND) [Bonet and Geffner, 2018, Bonet et al., 2019]; all these works are non-neural in nature. There is research [Toyer et al., 2018] on developing neural models over PPDDL, but since our focus is on RMDPs expressed in RDDL, and the architecture of neural reactive policies is tailored to the description language, these works are not directly comparable to ours. Issakkimuthu et al. [2018] learn Deep Reactive Policies for RDDL domains, however, their model is not capable of size transfer. We, instead, build upon a series of works [Bajpai et al., 2018, Garg et al., 2019, 2020], which proposes neural solvers for RDDL. Torpido [Bajpai et al., 2018] can only perform transfer on instances of same size, whereas TrapsNet [Garg et al., 2019] makes additional assumptions on the arities of state and action predicates. Closest to us is SymNet (Garg et al. [2020]), which, to our knowledge, is the only neural model for a general RDDL RMDP. We next describe its detailed architecture.

## 2.4 SYMNET

Given an RDDL domain and an instance $I$, SymNet (Garg et al. [2020]) solves TLR as follows: 1) first, represent $I$ in the form of an *instance-graph*, 2) use a GAT-based architecture to represent the generalized policy, 3) finally, train the model using a suitable end-to-end loss, e.g. RL-based or imitation learning based - we compare with both in our experiments. Next, we will discuss these steps in detail.

**Instance-Graph Construction:** We start by discussing how SymNet creates its instance-graph. In SymNet, the purpose of the instance-graph(s) is to translate an instance into graph(s) that capture interactions among various state-variables. For this, SymNet creates $|A| + 1$ graphs, $\mathcal{G}_{sym} = \{G_d, G_{a1}, \ldots, G_{a|A|}\}$. All graphs are derived from the DBN of the instance: $G_d$ captures exogenous, i.e. action-independent effects between state-variables, and each $G_{ai} \in \{G_{a1}, \ldots, G_{a|A|}\}$ captures effects between state-variables that are induced by action $ai$.

Recall that $O_{SF}$ represents the set of object tuples that appear in state-fluents. For each $o_{sf} \in O_{SF}$ SymNet adds a node $v$ with label $o_{sf}$ to each of the $|A| + 1$ graphs. Edges are introduced once all nodes are generated. In the following, let $v_1$ and $v_2$ be two nodes labeled with object tuples $o_1$, respectively, $o_2$. Whether an edge exists between $v_1$ and $v_2$ depends on the underlying graph: 1) for $G_d$ there is an edge between $v_1$ and $v_2$ if the DBN contains a state-variable $SP(o_1)$ that affects another state-variable $SP(o_2)$. Note that every state-variable affects itself, hence every node has a self-loop. 2) for $G_{ai} \in \{G_{a1}, \ldots, G_{a|A|}\}$ there

exists an edge between $v_1$ and $v_2$ if there is a state-variable $SP(o_1)$ and an action $a(o_a) \in A_{\mathcal{O}}$ of type $ai \in A$, that in conjunction affect another state-variable $SP(o_2)$. That is, it captures if a state-variable and some action of type $ai$ affect some other state-variable in the DBN.

**Node Features:** All graphs have the same set of input node features, determined by the following rules: a) For each parameterized predicate type $P \in SF$, a feature is added to every node $v$. For each grounding $P(o)$, the node feature of $o$ that corresponds to $P$ is set to the value of $P(o)$. The value is fetched from the current state. b) For each unparameterized Boolean non-fluent, a feature with its value is added to each node. c) A feature for a parameterized Boolean non-fluent is added to a node, if the object tuple corresponding to the non-fluent is a subset of the object-tuple at the node.

**Node Embeddings:** SymNet uses a Graph Attention Network (GAT) [Veličković et al., 2018], which is a specific kind of graph neural network that leverages the attention mechanism over a node's neighbors for its message passing updates. SymNet uses a GAT to compute node embeddings for each graph in $\mathcal{G}_{sym}$. We establish a correspondence between nodes in different graphs having the same label, i.e., which correspond to the same object tuple. A final node embedding $ne(v)$ for a node $v$ (representing all the nodes in different graphs having the same label) is constructed by: $ne(v) = concat(GAT_d(G_d)[v], ..., GAT_{a|A|}(G_{a|A|})[v])$. A global embedding $ge$ representing the complete state is then computed as a maxpool over all node embeddings as: $ge = maxpool_{v \in V}(ne(v))$ where $V$ is the set of all nodes.

**Action Decoding:** SymNet creates a set of action decoders $(AD_1, \ldots, AD_{|A|})$ for each action type in the domain. Let there be a parameterized ground action $a(o)$ that affects a set of state-variables $\mathcal{P}_{a(o)}$. Let $args(P)$ denote a function that returns the arguments of predicate $P$. Then, the score of action $a(o)$ is computed as $score(a(0)) = AD_{type(a)}(maxpool_{P \in \mathcal{P}_{a(o)}}(ne(args(P))), ge)$, where $type(a)$ returns the type of action $a$. To get a policy, $softmax$ is taken over all action scores.

## 3 SYMNET2.0: A NEW ARCHITECTURE

We formally discuss the shortcomings of SymNet's instance-graph and its architecture. We then propose SYMNET2.0 which overcomes these challenges by effective handling of non-fluents and actions in its architecture to learn a generalized neural policy.

### 3.1 RUNNING EXAMPLE

Recon is an IPPC domain where the agent moves in a 2D grid-world and is equipped with tools for detecting water, life, and taking pictures. Certain locations on the grid are marked as hazard and if the agent uses a tool on these

locations the tool gets damaged with a high probability. Once a tool is damaged the agent has to return to the base location where they can repair the tool. The agent is positively rewarded for taking pictures of cells where life is detected. The domain has:

**Objects Types:** `x,y,obj,agent,tool`.

**Non-Fluents:** `objAt(obj, x, y)`, `is_up(y₁,y₂)`, `is_down(y₁,y₂)`, `is_right(x₁,x₂)`, `is_left(x₁,x₂)`, `base(x, y)`, `hazard(x, y)`, `detect_prob_damaged`, `damage_prob(tool)`, `detect_prob`, `camera_tool(tool)`, `life_tool(tool)`, `water_tool(tool)`, `good_pic_weight`, `bad_pic_weight`.

**State-Fluents:** `agentAt(agent, x, y)`, `damaged(tool)`, `waterChecked(obj)`, `waterDetected(obj)`, `lifeChecked(obj)`, `lifeChecked2(obj)`, `lifeDetected(obj)`, `picTaken(obj)`.

**Actions**: `up(agent)`, `down(agent)`, `left(agent)`, `right(agent)`, `useToolOn(agent, tool, obj)`, `repair(agent, tool)`

We consider an instance with a $2 \times 2$ grid, where $\{x_1,x_2\}$ and $\{y_1,y_2\}$ are of type x, respectively y. There is one agent $ag_1$, two tools $\{t_1,t_2\}$, one object $\{o_1\}$ and `hazard(x₁, y₂)` and `objAt(o₁, x₂, y₁)` are `True`.

## 3.2 SHORTCOMINGS IN SYMNET

As motivated in Section 1, SymNet makes certain design choices which results in sub-optimal performance on several planning problems. First, since its instance graph is derived from the underlying DBN, it is incapable of capturing important information present in the RDDL description in the form of parameterized non-fluents. Specifically, SymNet's instance graph can only incorporate information about those non-fluents whose arguments also appear in a state-fluent; for all others, the information is compiled away. Second, the score of each action is decided solely on the basis of what state-variables the action affects. This means that any action arguments which do not appear in state-fluents affected by the action will have no impact on the action score, resulting in action non-identifiability as demonstrated by the following proposition. Given an action $a(o)$, we will use the notation $\mathcal{P}_{a(o)}$ to denote the set of state-variables (fluents) affected by $a(o)$.

**Proposition 1.** *Let there be two actions $a(o_1)$ and $a(o_2)$ of action type $type(a)$, where $o_1 \neq o_2$. Let both actions affect the same set of state-variables i.e. $\mathcal{P}_{a(o_1)} = \mathcal{P}_{a(o_2)}$. Then, the scores computed by SymNet for both of these actions will be identical.* [see Appendix for a proof]

In our example, non-fluent `objAt(obj,x,y)` indicates that the object `obj` is present at the location x, y, but since there is no state-fluent with this set of arguments, the grounding of this object tuple is never represented explicitly in the instance graph. Hence, the network may not generalize well to instances where objects are present at different locations than those seen during training. Further, there is an action `useToolOn(agent,tool,obj)` which says that `agent` uses `tool` on `obj`. Since this action only affects state fluents with object tuple `obj`, the embedding for `tool` is not incorporated during action decoding, resulting in an identical score for two actions applying different tools to the same object.

Because of above issues, SymNet results in learning suboptimal policies which do not transfer well to new instances for several domains. Next, we describe our approach which can handle these shortcomings in a comprehensive manner.

## 3.3 OUR APPROACH

To handle these shortcomings we will make two changes, 1) we add a set of new graphs to SymNet, and 2) we add new inputs to the action decoder. We explain these details next.

**Adding Position-based Graphs:** On top of graphs in SymNet, we create a new set of graphs $\{G_{p1}, \ldots, G_{p|Ar|}\}$ that capture what object comes at what position in a state-variable or non-fluent. Hence, we now have $\mathcal{G}_{sym2} = \{G_d, G_{a1} \ldots, G_{a|A|}, G_{p1}, \ldots, G_{p|Ar|}\}$, where $|Ar|$ is the maximum arity of any predicate in the domain.

Intuitively, these new graphs capture the relationship between object tuples in the instance, which could be part of a state-fluent or a non-fluent, and their individual object arguments. There is a different graph for each position that an argument could appear in, in order to capture the relative ordering of arguments. We next describe the set of nodes and edges for each of the graphs in $G_{sym2}$,

1) **Object Tuple Nodes:** For each $o_{sf} \in O_{SF}$ we add a vertex $u$ to each graph in $G_{sym2}$ with label $o_{sf}$. Note that these nodes are the same as those in SymNet's instance-graph. Similarly, for each $o_{nf} \in O_{NF}$ we add a vertex $v$ to each graph in $G_{sym2}$ with label $o_{nf}$. [3] These nodes are added to capture the missing information available in non-fluents which is not covered by SymNet.

2) **Singleton Object Nodes:** Finally, for each $\tilde{o} \in \mathcal{O}$ a vertex $w$ with label $\tilde{o}$ is added to each graph in $G_{sym2}$ (if it is not already added in the previous step). These new singleton object nodes are created for message passing to and from non-fluent based nodes. As a side benefit, we will see later that these singleton object nodes will also be helpful in removing action non-identifiability.

For each object-tuple $o \in O_{SF} \cup O_{NF}$, and for each object $o[i] \in \mathcal{O}$ appearing at position $i$ in $o$, we add edges $e(o, o[i])$ and $e(o[i], o)$ in $G_{pi}$. This means, each graph in $\{G_{p1}, \ldots, G_{p|Ar|}\}$ has bidirectional edges that capture

---

[3] In order to be memory efficient, we add these nodes only for non-fluents taking non-default value.

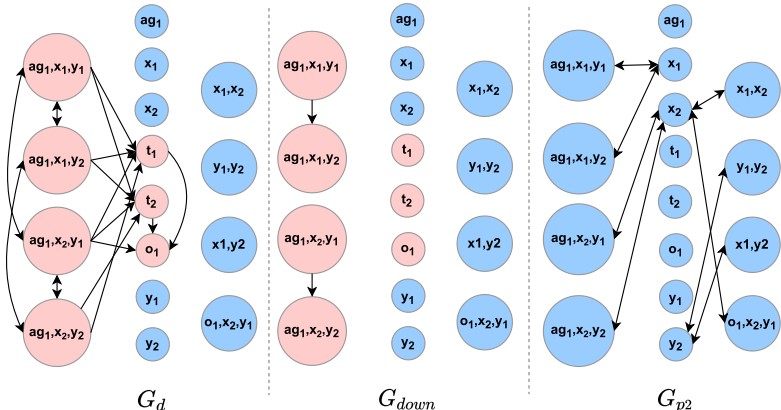

Figure 1: (left): Graph capturing action-independent effects (ref. 2.4), $G_d$; (middle): one of the six action induced graphs (ref. 2.4), $G_{down}$, for the `down` action; (right): one of the three position-based graphs (ref. 3.3), $G_{p2}$, for the second position. All nodes have a self loop (not shown for visual clarity). Red nodes are present in both SYMNET2.0 and SymNet, where as blue nodes are present only in SYMNET2.0. Position-based graphs, e.g., $G_{p2}$, are present only in SYMNET2.0.

whether an object occurs at position $i$ of any object-tuple (of any state-variable or non-fluent). Separate adjacency for each position is used to preserve ordering of objects in an object-tuple. This helps in preserving semantic meaning in predicates like `is_up(a,b)` where ordering of `a` and `b` matters, hence, `is_up(a,b)` and `is_up(b,a)` should be treated differently. Figure 1 shows the instance graphs of SymNet and SYMNET2.0 for our running example. We refer to the original paper of SymNet Garg et al. [2020] for construction of $G_d$ and $G_{down}$. $G_{p2}$ captures what objects appear as $2^{nd}$ argument of a state-fluent/non-fluent, e.g., $x_1$ is connected to $(ag_1, x_1, y_1)$ and $(ag_1, x_1, y_2)$.

**Node Features:** All newly constructed graphs have the same set of input node features, which are described as follows:

1) **State-Fluent Features:** For each parameterized state predicate type $P$, we add a feature to every node $v$. For each grounding $P(o)$ of $P$, the node feature of $o$ that corresponds to $P$ is set to the value of $P(o)$ fetched from the current state. For all other object tuples which do not appear as groundings of P this feature is set to the default value of $P$ from the domain file. We denote the set of the resulting features with $h^{SF}(v)$.

2) **Non-Fluent Features:** For each parameterized non-fluent predicate type $N$, we add a feature to every node $v$. For each grounding $N(o)$ of $N$, the node feature of $o$ that corresponds to $N$ is set to the value of $N(o)$. The value is fetched from the instance description for the latter, and from the domain description for the former. For all other object tuples which do not appear as groundings of $N$ this feature is set to the default value of $N$ from the domain file. We denote the set of the resulting features with $h^{NF}(v)$.

3) **Global Features:** Unparameterized state-fluents and non-fluents represent global properties relevant to all nodes, hence, these are added as features to every node. The values

are fetched from the current state for state-fluents and from the instance description for non-fluents. Let these features be denoted by $h^G(v)$.

4) **Type Features:** For each node $v$ with label $o$, we create a one-hot encoding vector $h^{TY}(v)$ representing the type of the node in the instance-graph(s). We define the type of each object-tuple $o = (o[1], \ldots, o[l])$ as $type(o) = (type(o[1]), \ldots, type(o[l]))$ where the type operator is overloaded to return the type of object given as input to it.

The overall node feature of a node $v$ is represented as:
$h(v) = concat(h^{SF}(v), h^{NF}(v), h^G(v), h^{TY}(v))$.

**Proposition 2.** *Let $u$ and $v$ be two nodes with label $o_u$ and $o_v$ corresponding to object tuples of some state-variables in $\mathcal{G}_{sym}$. Let $d_{sym}(u, v)$ denote the minimum distance between nodes $u$ and $v$ in any of the graphs in $\mathcal{G}_{sym}$ and let $d_{sym2}(u, v)$ denote the minimum distance between nodes $u$ and $v$ in any of the graphs in $\mathcal{G}_{sym2}$. Then, $d_{sym2}(u, v) \leq d_{sym}(u, v)$. [see Appendix for a proof]*

Proposition 2 shows that $\mathcal{G}_{sym2}$ can have shorter distances among nodes in the graph. This can result in better message passing as also demonstrated in Section 4.2.3.

**Node Embedding:** We use the similar GAT-based architecture as in SymNet to compute node embeddings for each graph in $\mathcal{G}_{sym2}$. Like in SymNet, we establish a correspondence between nodes in different graphs having the same label, i.e., which correspond to the same object tuple. A final node embedding $ne(v)$ for a node $v$ (representing all the nodes in different graphs having the same label) is constructed by: $ne(v) = mlp\big(concat(GAT_d(G_d)[v], \ldots, GAT_{a|A|}(G_{a|A|})[v], \ldots, GAT_{p|Ar|}(G_{p|Ar|})[v])\big)$. To represent the complete state, a global embedding $ge$ is then computed as a maxpool over all node embeddings as: $ge = maxpool_{v \in V}(ne(v))$, $V$

being the set of all nodes.

**Action Decoding:** To address the issue with SymNet's decoding, while computing the score of a parameterized action $a(o)$, we also give as input the node embeddings of each object occurring as a parameter in $a(o)$ along with the node embeddings of the nodes it affects. This leads to unique identification of each action as its parameters uniquely identify it. Formally, let there be a parameterized ground action $a(o)$ that affects a set of state-variables $\mathcal{P}_{a(o)}$ and let $o = (o[1], \ldots, o[n])$ then, the score $score(a(o))$ is given as: $AD_{type(a)}\big(ne(o[1]), \ldots, ne(o[n]),$ $maxpool_{P \in \mathcal{P}_{a(o)}}(ne(args(P))), ge\big)$. This implies that scores computed by SYMNET2.0 for two actions $a(o_1)$ and $a(o_2)$ with $o_1 \neq o_2$ and $P_{a(o_1)} = P_{a(o_2)}$ (ref. Proposition 1), will (in general) be different from each other (follows from the formula used for score computation).

## 3.4  TRAINING ALGORITHM

We use a two phase process to train SYMNET2.0 using imitation learning. In the first phase, referred to as dataset generation, for each training instance in the set of training instances $I_{tr}$ we use the PROST [Keller and Eyerich, 2012] planner, a state-of-the-art UCT-based *online* probabilistic planner, to generate a set of trajectories $\tau_1, \ldots, \tau_M$, where each trajectory is a sequence of state-action pairs $\langle s_0, a_0, \ldots, s_{H-1}, a_{H-1} \rangle$. To compute dataset $D_i$ we first compute the union of all state-action pairs among all trajectories. Since PROST is a sampling-based planner with time-limited lookahead, different trajectories can potentially contain state-action pairs $(s, a_i)$ and $(s, a_j)$, i.e. pairs which share the same state, but where a different action is applied. This may cause problems for the underlying neural learner. We circumvent this by only keeping the action which occurs most frequently for a given state and leave the exploration of other solutions for the future work.

In the second phase, referred to as neural learning, SYMNET2.0 is trained using supervised learning using the dataset generated in Phase 1 above. During training, we divide each $D_i$ into batches and we consume all batches of $D_i$ before moving to the dataset of the next instance. A cross-entropy based loss is used during training. During inference we take an $argmax$ over the action distribution to decide the action to be taken. Recall that the underlying GAT as well as the action decoder in SYMNET2.0 (and SymNet) share their respective parameters, making weight learning independent of a specific instance, and hence, these architectures seamlessly generalize to train/test instances of different sizes. We note that in the work done by Garg et al. [2020], SymNet was trained using an RL based loss. For a fair comparison, we experiment with SymNet using both kinds of losses, i.e., an RL based loss and imitation learning based loss, as described above.

## 3.5  REPRESENTATIONAL CAPABILITIES

SymNet is a special case of SYMNET2.0 in the following sense: (a) We set all the weights of GATs applied on the position-based Graphs ($\{G_{p1}, \ldots, G_{p|Ar|}\}$) to 0 rendering them inactive. We note that since there are no new edges added in the DBN-based graphs ($\{G_d, G_{a1}, \ldots G_{a|A|}\}$), any singleton nodes added in SYMNET2.0 do not participate in the message passing in these graphs. (b) We zero out the node embedding of any node which do not correspond to a node embedding for a state-fluent. Then, it is easy to see that the architecture SYMNET2.0 reduces to that of SymNet.

If the path length required for the propagation of relevant information required for learning an optimal policy is greater than the message passing depth then there is no possibility of finding such an optimal policy. Proposition 2 shows that SYMNET2.0, due to its architecture, never increases this required path length compared to SymNet. Hence, any policy which can be represented optimally by SymNet can also be represented by SYMNET2.0. However, the theoretical question that given a sufficient number of messaging passing steps, is it always possible for SYMNET2.0 to represent/learn the optimal policy for RDDL RMDPs, is still open and a direction for future work. Recently, Ståhlberg et al. [2022] concluded that generalized policies that can not be written in two-variable counting logic ($C_2$ logic) can not be represented/learned using Graph Neural Networks. Characterizing and finding RDDL domains where the optimal policy can be written in $C_2$ logic however is still an open problem to the best of our knowledge.

## 4  EXPERIMENTS

With our experiments, we want to answer three key questions. (1) IPPC performance: does SYMNET2.0 result in better performance on IPPC instances compared to SymNet? (2) how well does SYMNET2.0 generalize to instances that go far beyond the size of the largest IPPC instances, compared to other approaches? (3) how well does SYMNET2.0 generalize to instances where there is a significant difference between the non-fluents of the test instance and the non-fluents seen during training?

### 4.1  EXPERIMENTAL SETUP

**Domains:** We evaluate all models on twelve IPPC 2011 and 2014 domains: Academic Advising (Acad), Crossing Traffic (CT), Game of Life (GoL), Navigation (Nav), Skill Teaching (Skill), Sysadmin (Sys), Tamarisk (Tam), Traffic, Wildfire (Wild), Recon, Triangle Tireworld (TT) and Elevators (Elev) (ref. Appendix for domain descriptions). For each domain, we pick IPPC instances 1-3 as training instances, validate on instance 4 and test on instances 5-10 (unless stated otherwise). We validate on instance 4 by eval-

uating the checkpoints saved during training and picking the one with the best reward for final testing.

**Algorithms & Settings:** SymNet is the only published work for the task of training a generalized neural policy for RDDL RMDPs. It uses RL to train, which, in our preliminary experiments, suffers from exploration issues, due to the sparse rewards inherent to many IPPC domains. Since SYMNET2.0 is trained using imitation learning (IL), we create a stronger baseline by training the SymNet architecture also with the IL data. We name this system SymNet-IL. To construct IL data, for each training instance, we run PROST[4] in its default setting and collect 100 trajectories, which are converted to (state, action) pairs and used as IL training data.

SymNet is trained for 12 hours (as per original paper's setting). SYMNET2.0 and SymNet-IL are trained for 500 epochs with a maximum allowed training time of 12 hours (for parity). However, in practice, both IL-based models are much faster to train and take no more than 7 hours training (including data generation) in any domain.

We are guided by the literature on domain independent planning, where the goal is to develop a *single* planner that can work on any domain. So, we do not apply any domain specific hyperparameter tuning, and use a fixed neighborhood size of 1 in the GAT for all domains. Section 4.2.1 briefly discusses the effect of this hyperparameter.

Finally, we also compare against PROST. We emphasize that any direct comparison with PROST is not meaningful, as PROST is an online planner that uses interleaved planning and execution and the other three models are offline planners. Note that the neural (offline) planners require only a forward pass for each step of execution and hence are very fast during testing. In contrast, PROST is evaluated in its default setting on test instances. Nevertheless, we still include the comparison with PROST in terms of rewards obtained to gain a deeper insight into our results (generally, the expectation is that PROST will perform better as it can perform target interleaved exploration for the states that are actually reached). This implies that at test time it will be slower than the other approaches, but its overall training plus test time can still be lower. We do not report comparison of running times due to the aforementioned reasons.

**Evaluation Metric:** We follow existing literature on neural MDP solvers [Bajpai et al., 2018, Garg et al., 2019, 2020] and use the evaluation metric ($\alpha$) that outputs a number between 0 and 1, with 0 denoting a performance equal to random, and 1 denoting the best reward amongst all comparison approaches. In more detail, for a given domain, we run the train-validate-test cycle 3 times for each model $m$ (neural models, PROST, and random policy). For the $r^{th}$ run of $m$, we execute its policy for 200 episodes on each test instance $i$, and store the average long term reward as

$V(m, i, r)$. The maximum value of $V(m, i, r)$ is denoted as $V_{max}(i)$, and $V_{rand}(i)$ is the long term reward of the random policy.

Next, we assess the relative performance of a policy by computing a normalized metric $\alpha(m, i, r) = \frac{V(m,i,r)-V_{rand}(i)}{V_{max}(i)-V_{rand}(i)}$. To estimate the performance of a model $m$ on a domain, we compute $\alpha(m) = \frac{1}{|r|}\sum_r \frac{1}{|i|}\sum_i \alpha(m, i, r)$. If this metric is 1, that means that it outputs the best score in every instance. A negative value denotes that it outputs worse than random policies on average.

## 4.2   RESULTS

Table 1 reports our main result – all models tested on 12 IPPC domains in the setting described above. Each $(m, d)^{th}$ entry represents $\alpha(m)$: the performance of algorithm $m$ on domain $d$. The last column shows the mean over all 12 domains. Results of PROST are in gray color, as those numbers are not suitable for a direct comparison, but give a deeper insight into the overall performance quality. The bold values show the neural model with maximum $\alpha(m)$.

Overall, SYMNET2.0 outperforms SymNet-IL and RL based SymNet by vast margins of +22 and +36 points, respectively. In particular, SYMNET2.0 is better than the improved baseline SymNet-IL in 10 out 12 IPPC domains, and very close in the eleventh (TT). SymNet-IL gets superior results compared to SymNet, underscoring the difficulty in RL based training, and the value of imitation learning. Another noteworthy point is that in no domain is SYMNET2.0's performance close to or worse than random (see Recon and Skill for comparison with SymNet-IL), suggesting that the new instance graph with a better treatment of non-fluents improves the overall model generalization. A paired T-test[5] comparing the mean rewards across 72 instances (12 domains with 6 test instances each) shows that our gain over SymNet is statistically significant with a $p$-value of 0.9994 (see Appendix for details).

### 4.2.1   Ablation on Neighborhood Size

We determine the influence of neighborhood size of the GAT, by varying this hyperparameter from 1 to 3. For both Symnet-IL and SYMNET2.0, increasing the neighborhood size to 2 increases the performance in some domains (TT, Acad, Elev, Skill and Recon), but decreases performance in others, causing an overall decrease in performance. For best performance on a domain, this hyperparameter tuning could be easily done on the validation instance. Detailed results are available in the Appendix in Table 2. For the remainder, unless otherwise stated, we set this parameter to 1.

---

[4]https://github.com/prost-planner/prost

[5]https://docs.scipy.org/doc/scipy/reference/generated/ scipy.stats.ttest_rel.html

| IPPC Test Instances 5-10 | | | | | | | | | | | | | |
|---|---|---|---|---|---|---|---|---|---|---|---|---|---|
| **Model** | **TT** | **CT** | **Acad** | **Elev** | **Tam** | **Nav** | **GoL** | **Skill** | **Sys** | **Wild** | **Traffic** | **Recon** | **Mean** |
| PROST | 0.53 | 0.86 | 0.47 | 1.00 | 0.94 | 0.88 | 1.00 | 1.00 | 0.65 | 0.70 | 1.00 | 0.99 | 0.84 |
| SymNet | 0.00 | 0.37 | 0.58 | 0.31 | 0.55 | 0.53 | 0.20 | -0.40 | 0.62 | 0.27 | 0.00 | 0.03 | 0.26 |
| SymNet-IL | **0.83** | 0.91 | 0.72 | 0.38 | 0.63 | **0.56** | 0.20 | -0.50 | 0.49 | 0.72 | -0.18 | 0.03 | 0.40 |
| SYMNET2.0 | 0.81 | **0.95** | **0.82** | **0.44** | **0.92** | 0.47 | **0.29** | **0.43** | **0.94** | **0.77** | **0.28** | **0.30** | **0.62** |
| Larger Instances | | | | | | | | | | | | | |
| **Model** | **TT** | **CT** | **Acad** | **Elev** | **Tam** | **Nav** | **GoL** | **Skill** | **Sys** | **Wild** | **Traffic** | **Recon** | **Mean** |
| PROST | 0.09 | 0.55 | 0.39 | 1.00 | 0.90 | 0.44 | 0.91 | 1.00 | 0.36 | 1.00 | 1.00 | 0.78 | 0.70 |
| SymNet | 0.00 | 0.14 | 0.60 | 0.15 | 0.43 | 0.41 | 0.60 | -0.82 | **0.51** | 0.09 | 0.25 | 0.02 | 0.20 |
| SymNet-IL | **0.96** | 0.62 | 0.63 | **0.22** | 0.52 | 0.19 | 0.25 | -0.79 | -0.65 | **0.22** | 0.03 | 0.02 | 0.19 |
| SYMNET2.0 | 0.95 | **0.89** | **0.77** | 0.19 | **0.94** | **0.95** | **0.84** | **0.34** | 0.46 | 0.20 | **0.39** | **0.32** | **0.60** |

Table 1: Comparison between SYMNET2.0 and the baselines on 12 IPPC domains. All models are trained on (smaller) instances 1-3 and validated on instance 4. Upper part shows results on IPPC test instances 5-10 and lower part shows results on much larger instances than those in the IPPC. Bold values show the best performer among all neural models.

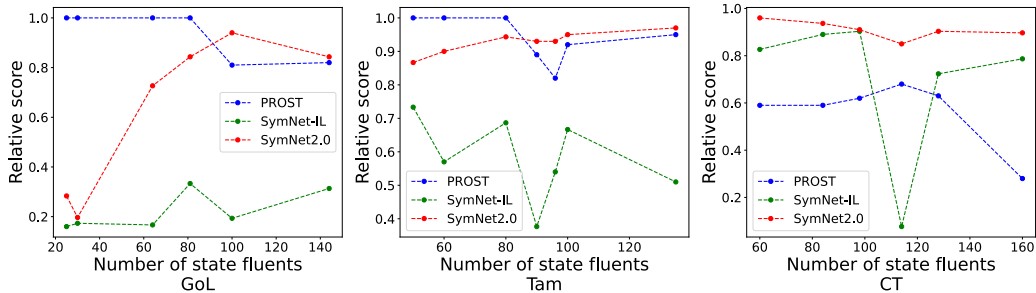

Figure 2: Performance trends on instances of increasing size: PROST deteriorates, but SYMNET2.0 remains robust.

### 4.2.2 Offline vs. Online Planning on Larger Instances

When comparing results of the online planner (PROST) with SYMNET2.0, we find that, overall, generalized neural policies are not able to match up to interleaved planning and execution. This is not entirely surprising, since the latter can target exploration based on specific observed outcomes of actions taken earlier. However, interestingly, we find that in a few domains (e.g., TT, Acad), SYMNET2.0 is able to outperform PROST. We hypothesize that this could be due to SYMNET2.0's ability to generalize well to large instances.

To test this hypothesis, we create four new test instances[6] for each domain (we call them instances 11 to 14), with sizes much larger than IPPC instances.[7] For some of the domains our instance#14 has three times the number of objects of IPPC's instance#10. For example, TT instance#10 has 66 grid cells, where our instance#14 has 190. Similarly, Acad instance#10 has 30 courses, where our instance#14 has 90. See Appendix for details on exact sizes. Additionally, we

increase the horizon to 100 for these larger instances.

Table 1 shows the comparison. We first notice that the gap between SymNet-IL and SYMNET2.0 increases drastically, when tested on larger instances (compared to previous experimental setting). This suggests that SYMNET2.0 generalizes more robustly to large problem sizes. We then compare the same gaps between PROST and SYMNET2.0, and find that, in aggregate, SYMNET2.0 closes in on PROST, and reduces the performance gap. In 8 of 12 domains (TT, CT, Acad, Tam, Nav, GoL, Traffic, Recon) the gap is reduced, whereas it gets worse in only 4 domains.

Figure 2 shows that PROST's relative performance starts to drop, as size increases. Two interesting cases are GoL and Tam, where in aggregate SYMNET2.0 performs worse than PROST, but in the figure, we observe that for the largest instances (13 and 14), it starts to outperform PROST. We conjecture that the reason for such results is that larger instances have larger state spaces, branching factors and reward horizon, due to which UCT based online planners like PROST may struggle to find high reward trajectories. In such scenarios, the size-invariance of generalized neural policies makes their additional benefit even more evident.

---

[6]We will release these instance files for further research.

[7]generated using the official scripts provided by the IPPC at https://github.com/ssanner/rddlsim

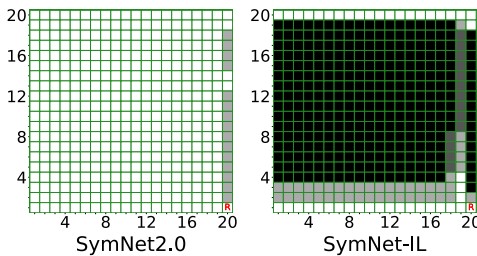

Figure 3: Coverage of SYMNET2.0 (left) and SymNet-IL (right) on grid size $20 \times 20$ when trained on grid size $5 \times 5$.

### 4.2.3 Generalization to Changing Non-fluents

Non-fluents of a domain control the underlying structure and parameters that affect the transition model and are critical for finding a good policy for a given instance. The non-fluent values vary from instance to instance, and hence it is important for a generalized policy to be robust to these changes. In most IPPC domains, these non-fluents vary considerably and hence our results in Table 1 already provide some evidence for our model's ability to adapt. However, we hypothesize that the gains should not be attributed only to a better non-fluent handling, but also to the newly added singleton nodes. We believe that these singletons facilitate better localization and sharing of information.

To verify this, we create a simple variation of the Navigation domain (without action stochasticity) and vary the goal non-fluents. Similar to a regular Navigation domain, the robot always starts at the lower right corner of a 2D-grid and has to reach a goal using five actions: North, South, East, West and *noop*. It gets a reward of 0 on reaching the goal and -1 otherwise. A state-fluent robotAt(x,y) and a non-fluent goalAt(x,y) specify the locations of robot and goal respectively. In IPPC instances, the goal non-fluent is always at the upper right corner. However, in our experiment, we test the model by marking each grid cell as the goal in turn – essentially checking the model's ability to learn to solve simple path planning problems.

We train SymNet-IL and SYMNET2.0 on instances of size $5 \times 5$. The dataset for this experiment was generated using a human policy rather than PROST. To factor out any lack of diversity, we create 24 training instances, one for each grid cell as a goal. For validation we create three instances of size $11 \times 11$ where the goal is kept at locations $(4, 4)$, $(4, 7)$, and $(5, 5)$ (ref. Figure 3) and the model with the best average reward on these is selected. For testing, a total of 399 instances of size $20 \times 20$ are used. In Figure 3, we report the fraction of test instances for both the models where the robot is able to reach the goal averaged over three different runs. Each cell has one of the four colors: black, dark grey, light grey and white, denoting the coverage ratios of 0/3, 1/3, 2/3 and 3/3, respectively, for instances where the goal is located at that cell. Clearly, the coverage for SYMNET2.0

is enormously higher than for SymNet-IL.

Further analysis reveals that the instance graphs of both models already incorporate the knowledge of goal(x,y) as a feature in node (x,y). Hence, the better coverage of SYMNET2.0 cannot be due to a better handling of non-fluents. The main difference in the two graphs is the addition of singleton nodes and corresponding edges between object tuple nodes (x,y) in the position based graphs in $\mathcal{G}_{sym2}$. We believe that these singleton nodes lead to better information exchange among nodes. Nodes x and y can act as representatives of rows and columns: if the goal is at location (x,y), then the node x could learn features like robotAt(x,*) $\land$ goalAt(x,*) (* represents don't care), i.e., a feature that signifies whether the robot is in the same column (analogously row) as the goal. In case of SymNet, singleton nodes are absent, hence it requires message passing steps of arbitrary length to localize the goal, thus, hurting its generalizability.

## 5 CONCLUSION

We present SYMNET2.0, a neural architecture for learning generalized policies for relational MDP domains expressed in RDDL. Its key technical contribution is a better handling of non-fluents by creating nodes for object tuples that occur as arguments to a non-fluent. It also creates singleton object nodes, when not present, and uses these in the action decoder, which mitigates the problem of action non-identifiability in the previous SymNet system. Extensive experiments reveal that not only is SYMNET2.0 vastly superior to SymNet, it is also more robust to large instance sizes, and generalizes well with changing non-fluents. Directions for future work include combining PROST with SYMNET2.0, and extending it to other settings such as Concurrent MDPs [Mausam and Weld, 2004] and POMDPs.

### Acknowledgements

Vishal Sharma is supported by TCS Research Scholar Fellowship. Mausam and Parag Singla are/were supported by IBM SUR awards, and Visvesvaraya Young Faculty Fellowship by Govt. of India. Mausam is supported by grants from Huawei, Google, Bloomberg, and a Jai Gupta Chair Fellowship. Parag Singla was supported by the DARPA Explainable Artificial Intelligence (XAI) Program #N66001-17-2-4032. We thank IIT Delhi HPC facility[8] for computational resources. We thank Gobind Singh and Siddhant Mago for discussions during the initial phase. Any opinions, findings, conclusions or recommendations expressed in this paper are those of the authors and do not necessarily reflect the views or official policies, either expressed or implied, of the funding agencies.

---

[8]*https://supercomputing.iitd.ac.in*

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
