# OpenReview forum: "SymNet 2.0: Effectively handling Non-Fluents and Actions in Generalized Neural Policies for RDDL Relational MDPs"
_auai.org/UAI/2022/Conference — UAI 2022 Poster_

### Official Review · Reviewer_6NKG · 2022-04-05

**Q2(1) Originality/Novelty:** 3
**Q2(2) Significance/Impact:** 3
**Q2(3) Correctness/Technical Quality:** 3
**Q2(6) Clarity Of Writing:** 3
**Q6 Overall Score:** 6
**Q8 Confidence In Your Score:** 3

**Q1 Summary And Contributions:**

The paper proposes "Effectively handling Non-Fluents and Actions Network" (ENFANet) for solving relational MDPs (RMDPs). ENFANet builds upon SymNet, building a denser graph and using a richer decoder. The experimental results demonstrate improvements.


**Q10 Ethical Concerns (Optional):**

I do not see and ethical concerns that go beyond ethical concerns that one may have about automating intelligent behavior.

**Q2 Assessment Of The Paper:**

More detailed information regarding each of these aspects is given below:

**Q2(4) Quality Of Experiments (Optional):**

3: Good: The experimental evaluation is adequate, and the results convincingly support the main claims.

**Q2(5) Reproducibility:**

3: Good: Key resources (e.g., proofs, code, data) are available and key details (e.g., proofs, experimental setup) are sufficiently well-described for competent researchers to confidently reproduce the main results.

**Q3 Main Strengths:**

The core insight is the more principled handling of non-fluents within SymNet. Specifically the paper should present the novel encoding and then show that SymNet is a specialized version. This is interesting and shows a lot of benefits.

**Q4 Main Weakness:**

While the paper focuses on deep neural approaches, it would be good to cite some more approaches that are not neural. It is critical to highlight some of the assumptions such as the transfer setting. For instance, traditional boosting approaches to solving RMDPs do not need this. Anyhow, this is really a minor issue.

The main downside is the presentation. The paper is very much presented in a SymNet-centric way. In turn, this reads very much like an engineering exercise. This, however, is not the case, in my opinion, see Q3

The experimental results support the effectiveness of ENFANet, though the ablation study on neighborhood size should rather be a scaling experiment?

Though the experiments show what they are supposed to show, I am still wondering whether one could actually present results where SymNet would essentially break down?

**Q5 Detailed Comments To The Authors:**

The paper proposes "Effectively handling Non-Fluents and Actions Network" (ENFANet) for solving relational MDPs (RMDPs). ENFANet builds upon SymNet, building a denser graph and using a richer decoder. The experimental results demonstrate improvements.

While the paper focuses on deep neural approaches, it would be good to cite some more approaches that are not neural. This is critical to highlight some of the assumption such as the transfer setting. For instance, traditional boosting approaches to solving RMDPS do not need this. Anyhow, this is really a minor issues.

The main downside is the presentation. The paper is very much presented in a SymNet-centric way. In turn, this reads vey much like an engineering exercises. This, however, is not the case, in my opinion. The core insight is the more principled handling of non-fluents. This should be the center for the paper. In other words, the paper should present the novel encoding and then show that SymNet is a specialized version. Then also the differences in the learning approach should be highlighted.
The experimental results support the effectiveness of ENFANet, though the ablation study on neiughbourhood size should rather be a scaling experiments? Though the experiments show what they are suppose to show, I am still wondering whether one could actually present results where SymNet would essentially break down?

To summarize, simple but very effective extension of SymNet for solving RMDPs.  While the paper could have been presented in stronger way, the contributions are interesting.

**Q7 Justification For Your Score:**

I like the modification of SymNet. And the changes provide improvements.

**Q9 Complying With Reviewing Instructions:**

1: Yes.

---

### Official Review · Reviewer_W9Zt · 2022-04-10

**Q2(1) Originality/Novelty:** 3
**Q2(2) Significance/Impact:** 3
**Q2(3) Correctness/Technical Quality:** 4
**Q2(6) Clarity Of Writing:** 4
**Q6 Overall Score:** 7
**Q8 Confidence In Your Score:** 4

**Q1 Summary And Contributions:**

The paper introduces a new graph neural network called ENFANet to represent and solve generalized policies for relational MDPs. The key idea is to enrich the previous architecture called SymNet by adding nodes for non-fluent tuples and for objects that occur in those tuples. This addresses the representational weakness of SymNet and makes it possible to learn effective and general policies for a number of benchmark domains.

**Q10 Ethical Concerns (Optional):**

No.

**Q2 Assessment Of The Paper:**

More detailed information regarding each of these aspects is given below:

**Q2(4) Quality Of Experiments (Optional):**

4: Excellent: The experimental evaluation is comprehensive and the results are compelling.

**Q2(5) Reproducibility:**

3: Good: Key resources (e.g., proofs, code, data) are available and key details (e.g., proofs, experimental setup) are sufficiently well-described for competent researchers to confidently reproduce the main results.

**Q3 Main Strengths:**

The paper attacks an important problem, namely of learning general (size-independent) policies for relational MDPs.

The approach addresses the representational inadequacy of SymNet by adding nonFluents and objects as nodes.

The experiments in a number of domains show compelling results that it works better than SymNet in both RL and imitation modes. The results are comparable or better than PROST when the problem size is increased.

The paper is well written and well-justified by the theory.

**Q4 Main Weakness:**

The theory is not as strong in certain aspects. For example, it would be nice to know if the network can represent any finite horizon optimal policy after a finite number of steps of propagation of activations. The current results show that the distance between nodes is decreased for the new network, which is to be expected when new nodes and edges are added to the network. It does not show that the current architecture does not also have another representational inadequacy similar to that shown for SymNet.




**Q5 Detailed Comments To The Authors:**

The paper badly needs diagrams of instance graphs in SymNet and ENFANET to clarify the representations and help make your points.



Section 3.4 that describes training and inferencc is too brief. Since the network size depends on the problem size, it is not clear how size-independent weights are learned. How are the activations propagated in the network during inference? All this needs more detail.

Appendix is too elaborate and can be shortened.

**Q7 Justification For Your Score:**

Learning size-independent policies for relational MDPs is a hard open problem. The results here are fairly compelling on multiple relational MDPs. The paper fixes the problems with SymNet using a principled representation. It appears that a stronger theoretical results is provable, but even without it, it is a nice paper.

**Q9 Complying With Reviewing Instructions:**

1: Yes.

---

### Official Review · Reviewer_PBDn · 2022-04-13

**Q2(1) Originality/Novelty:** 2
**Q2(2) Significance/Impact:** 2
**Q2(3) Correctness/Technical Quality:** 3
**Q2(6) Clarity Of Writing:** 3
**Q6 Overall Score:** 5
**Q8 Confidence In Your Score:** 4

**Q1 Summary And Contributions:**

The paper proposes a graph neural network extending that of SymNet to include parts of the relational MDP description that is invisible to SymNet, and demonstrates improved scalability on relational MDP problems.

**Q2 Assessment Of The Paper:**

More detailed information regarding each of these aspects is given below:

**Q2(4) Quality Of Experiments (Optional):**

3: Good: The experimental evaluation is adequate, and the results convincingly support the main claims.

**Q2(5) Reproducibility:**

3: Good: Key resources (e.g., proofs, code, data) are available and key details (e.g., proofs, experimental setup) are sufficiently well-described for competent researchers to confidently reproduce the main results.

**Q3 Main Strengths:**

The work identifies factors in the scalability of GNN based solution methods for relational MDPs, and develops methods to utilize them.

**Q4 Main Weakness:**

The main focus is in state variables that no actions change but that still determine the system behavior, and which are not available for SymNet but are available to ENFANet. This is an interesting observation, but this issue and its solutions do not appear to be very profound.

**Q5 Detailed Comments To The Authors:**

You characterize PROST as a "state-of-the-art online planner" for MDPs. The area of MDPs is quite broad, and it would useful to characterize this more accurately. My guess is that you are talking about MDP methods that have been tested on the IPPC problems.

You could explain the experiment setup and the evaluation metric alpha that is used in Table 1 a little bit deeper. If I am guessing correctly, the neural MDP solvers generate an action sequence very quickly, so there is no question about how high the runtimes are, and in addition to learning effort, only the rewards the action sequences produce are interesting to compare.

The higher mean alpha for ENFANet is due to the two cases in which SymNet produces a negative alpha, indicating much worse behavior than random. It would be interesting to explain this a bit deeper. In many cases the differences seem small or insignificant.


**Q7 Justification For Your Score:**

The work points out a weakness in SymNet's learning ability, but this weakness is not very fundamental, and its fix is quite natural.

**Q9 Complying With Reviewing Instructions:**

1: Yes.

---

### Decision · Program_Chairs · 2022-05-15

**Decision:**

Accept (Poster)

**Comment:**

Meta Review: The reviewers agreed that this paper makes a valuable contribution. There are many suggestions in the reviews (and your response) that should be taken into account when revising this paper.